# Study of the Hydrodynamic Unsteady Flow Inside a Centrifugal Fan and Its Downstream Pipe Using Detached Eddy Simulation

**Jian-Cheng Cai [1,2], Hao-Jie Chen [1], Volodymyr Brazhenko [1,2] and Yi-Hong Gu [3,\*]**

1   College of Engineering, Zhejiang Normal University, Jinhua 321004, China;
    cai_jiancheng@foxmail.com (J.-C.C.); howchieh@163.com (H.-J.C.); v_brazhenko@ukr.net (V.B.)
2   Key Laboratory of Urban Rail Transit Intelligent Operation and Maintenance Technology & Equipment of
    Zhejiang Province, Jinhua 321005, China
3   School of Mechatronic Engineering, Quzhou College of Technology, Quzhou 324000, China
\*   Correspondence: gyhtianbao@yeah.net

**Abstract:** The detailed unsteady turbulent flow inside a centrifugal fan and its downstream pipe was studied using detached eddy simulation (DES) at three flowrates, namely, the best efficiency point (BEP), 0.75BEP, and 1.49BEP. Both the mean and fluctuating flow fields were analyzed on the basis of the root-mean-square value as the indication of fluctuating intensity. Results showed that the pressure fluctuation had the minimum value at BEP, but the velocity fluctuation increased with the flowrate. Most regions inside the centrifugal fan underwent large pressure fluctuation with the magnitude of about 10~20% of $p_{ref} = 0.5 \, \rho u_2^2$, where $u_2$ is the blade velocity at the impeller outlet. The pressure fluctuation had a maximum value at the impeller side of the tongue tip rather than the stagnation point, and it decreased rapidly along the outlet pipe with magnitude about 1% of $p_{ref}$ after distance of five pipe diameters. The spectra of hydrodynamic pressure showed conspicuous spikes at the blade passing frequency (BPF) in the volute but not in the downstream pipe. At the downstream pipe entrance, pressure fluctuation spectra agreed with experimental results, showing that hydrodynamic pressure fluctuations were dominant; however, the experimental data showed a much slower decreasing rate due to the acoustic fluctuations.

**Keywords:** centrifugal fan; unsteady flow; hydrodynamic pressure fluctuation; detached eddy simulation

## 1. Introduction

Fans, as typical fluid machinery devices, are widely employed for industrial and civilian usage; for transportation of working fluids in industry; for ventilation in buildings, tunnels, and cars; and for cooling in electric and electronic devices [1]. Since fans are ubiquitous, they consume a huge amount of electricity, which is usually generated by fossil fuel. Improvement of fan efficiency can reduce the energy consumption so that it is helpful to environmental sustainability. Detailed investigation of the internal unsteady flows of fans is the first step to improve the fan performance and reduce fan noise.

Fans can be classified as either axial, centrifugal, or mixed flow types. Centrifugal fans can achieve high pressure rise in a short axial distance compared with axial fans due to the fact that the work done by the impeller is through the centrifugal flow. For centrifugal fans, the internal flow of the impeller can be very complex due to the influence of the centrifugal and Coriolis forces as well as the pressure gradients. The impeller rotation causes pressure fluctuations, which can lead to exciting the structures, such as the vibrations of the volute casing and the pipeline, as well as aerodynamic noise producing. Therefore, it is important and necessary to study the internal flow fields including both the mean flow and the fluctuating flow for long-term safe and quiet operation.

The flow fluctuations inside a centrifugal fan are mainly induced by the periodic interaction between the rotating impeller and the stationary casing, especially the volute

tongue. The blade passing frequency (BPF) fluctuation induced by blades sweeping the volute tongue is usually the predominant component in the spectra. There are also broadband fluctuations due to the turbulent flow fluctuations in the inlet stream and boundary layers over structure surfaces.

In an earlier experimental study, Kjork and Lofdahl [2] carried out the measurements of the three mean velocity components and the Reynolds stresses values inside a centrifugal fan impeller. They found that the mean flow can be characterized as an attached flow with almost linearly distributed velocity profiles, and for the turbulent flow, turbulent stresses with relatively low values predominate in the center region of the channel. Velarde-Suarez et al. [3] experimentally investigated the flow at exit radial locations of a centrifugal fan with forward-curved blades using hot wire anemometry techniques and found a strong flow asymmetry with considerable changes in both magnitude and direction along the different circumferential positions, especially in circumferential positions closer to the volute tongue. Recently, Zhang et al. [4] investigated the casing static pressure distribution of a centrifugal compressor ranging from the choke to stall operating conditions in order to find the relationship between the casing pressure evolution and stall behavior under different rotational speeds. Ni et al. [5] studied the unsteady pressure and velocity fluctuations using laser doppler velocimeter in pump and discussed the relation between the unsteady pulsation and flow structure.

Computational fluid dynamics (CFD) has been a powerful tool to simulate the unsteady 3D flowfields of turbomachinery. Ballesteros-Tajadura et al. [6] carried out a simulation of the complete 3D unsteady flow inside an centrifugal fan, and analyzed the pressure fluctuations in some locations over the volute wall, showing an important BPF peak. Younsi et al. [7] solved the unsteady Reynolds-averaged Navier–Stokes equations (URANS) with scale adaptive simulation (SAS) as the turbulence modelling to study the overall performances of a centrifugal fan and the wall pressure fluctuations upon the volute casing surface. Spence and Amaral-Teixeira [8] performed a CFD parametric study of the effect of geometrical variations on the pressure pulsations and performance characteristics of a centrifugal pump, finding that the cutwater gap and vane arrangement had the greatest influence across the various monitored locations and the flow range. Jing et al. [9] performed simulation using the URANS approach with the shear stress transport (SST) $k-\omega$ turbulence model to study the interactions between the non-uniform impeller flow and the fixed volute through which the significant pressure fluctuations arise. Gao et al. [10] studied the unsteady flow in a large centrifugal pump with stay vanes on the basis of numerical and experimental approaches. Yao et al. [11] used detached eddy simulations (DES) [12] to investigate the influence of wall roughness on the static performance and pressure fluctuations of a double-suction centrifugal pump. Bozorgasareh et al. [13] designed a new impeller configuration with innovative shrouds, studying its influence on the pressure head and efficiency of the centrifugal pump from both experimental and numerical approaches.

There are only a few works studying the unsteady flow fields in the downstream pipe of a centrifugal pump or fan, although in many applications, pumps and fans are installed in the pipeline. At the same time, many papers concerned pipe flow exclusively, and the 90° pipe bends in particular can be found. Guala et al. [14] and Discetti et al. [15] studied the large-scale and very large scale motions in turbulent pipe flow using particle image velocimetry (PIV) measurement. Tan et al. [16] performed large eddy simulation (LES) combined with a characteristic-based split scheme to study the flow in two circular sections of 90° pipe bends, observing an additional pair of vortexes near the curved section of the pipe inner side. Dutta et al. [17] numerically studied flow separation in 90° pipe bend with high Reynolds number by $k-\varepsilon$ turbulence modelling. Röhrig et al. [18] made a comparative computational study of turbulent flow in a 90° pipe elbow, demonstrating the superiority of LES over the RANS approaches; however, this was at the cost of significant increase in computational effort. Zimoń et al. [19] computed the mixed convection flow through a u-shaped bend with the transient thermal boundary conditions applied at the inlet.

Regarding this fact, in this work, the full 3D unsteady turbulent flow at three flow rates inside a centrifugal fan and its downstream pipe was simulated though CFD calculations by DES. The detailed time-averaged and fluctuating flow fields were analyzed, especially the characteristics of the hydrodynamic pressure fluctuations in the downstream pipe at different flowrates. DES approach was employed to address the challenge of internal unsteady flow with high Reynolds number and massive flow separations. DES combines LES and RANS, spurred by the belief that each alone was powerless to solve the problem at hand [20].

Although LES showed good performance in capturing unsteady flow features, the computational cost is much higher than in industrial widely adopted unsteady RANS models [21]. For the no-slip condition on solid walls, the LES-filtered N-S equations are integrated to the wall, which requires fine cells with near-wall cell points $y^+ \leq 1$. Moreover, the resolution needs in the outer region of the boundary layer are very high, i.e., with at the least 20 points per thickness $\delta$ in each direction [20]. In the present study, the centrifugal fan had many wall surfaces, which leads to enormous computational cost of pure LES; therefore, the DES, specifically the delayed DES (DDES) [22], was employed here. Recently, Zhang et al. [23,24] applied DDES on the basis of SST $k-\omega$ as RANS approach to study the unsteady flow in a centrifugal pump, and good agreement with the PIV experimental measurement results was found.

## 2. Materials and Methods

### 2.1. Description of the Centrifugal Fan

The main dimensions of the studying centrifugal fan are shown in Table 1. The impeller with the outlet diameter $d$ = 400 mm had $Z_b$ = 12 forward-curved blades, and a vaneless rotating diffuser with the diameter $d_2$ = 460 mm. The rotational speed was $N$ = 2900 rpm; therefore, the rotational frequency was RF = $N/60$ = 48.3 Hz and the blade passing frequency was BPF = RF $\times$ $Z_b$ = 580 Hz. The blade velocity at the impeller outlet was $u_2 = 2\pi N/60 \times d/2$ = 60.73 m/s, and the reference dynamic pressure was $p_{\text{ref}}$ = 0.5 $\rho u_2{}^2$ = 2259.5 Pa.

**Table 1.** The main dimensions of the centrifugal fan.

| Parameters | Value |
|---|---|
| Impeller outlet diameter, $d_2$ (m) | 0.400 |
| Impeller outlet width, $b_2$ (m) | 0.036 |
| Diffusor outlet t diameter, $d_3$ (m) | 0.460 |
| Impeller inlet diameter, $d_1$ (m) | 0.155 |
| Impeller inlet width, $b_1$ (m) | 0.700 |
| Blade outlet angle, β$_2$ (°) | 126 |
| Blade inlet angle, β$_1$ (°) | 38 |
| Radius to tongue (m) | 0.246 |
| Number of blades, $Z_b$ | 12 |
| Rotational speed, $N$ (rpm) | 2900 |

The airflow leaving from the impeller gathered in the volute housing and then entered the downstream pipe. The fan outlet had a rectangular section and was connected by a transitional pipe piece (length of 0.245 m), which gradually changed from the rectangular section to a circular section with a diameter of 190 mm. A pipe of 7 m long was connected to the circular end of the transitional pipe piece, and thus the pressure fluctuations in the downstream pipe could be investigated.

On the basis of standard ISO 5801:2007 industrial fans performance testing using standardized airways, we tested the fan performance in a standard test rig that had the uncertainty within 2%. The non-dimensional fan performance with the pressure rise and efficiency is shown in Figure 1. The flow coefficient was determined by $\varphi = Q/(u_2\pi d_2^2/4)$, where $Q$ is volume flowrate. The pressure coefficient was found by $\psi = \Delta p_{tot}/p_{ref}\psi =$

$\Delta p_{tot}/p_{ref}$, where $\Delta p_{tot}$ is the total pressure rises. The BEP flow rate falls in 1000~1200 m$^3$/h. In this study, 1116 m$^3$/h was taken as the BEP flowrate, i.e., $\varphi$ = 0.0406.

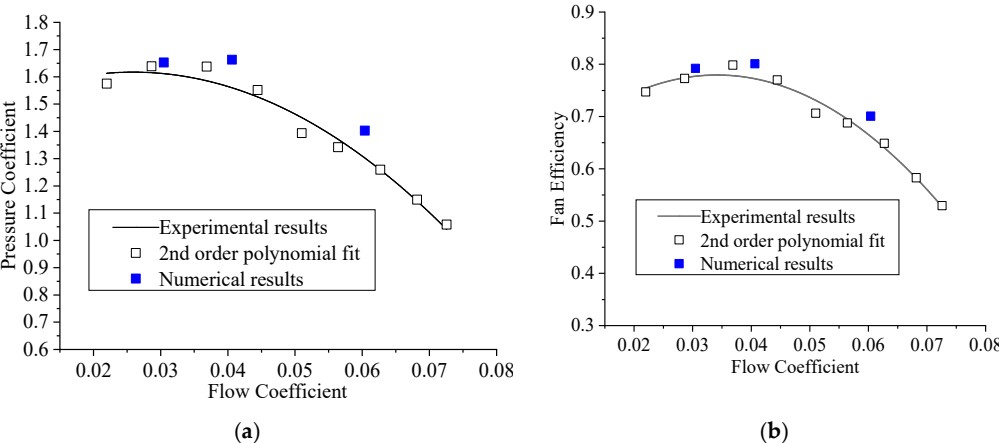

(a)                                                                                  (b)

**Figure 1.** Fan performance: (**a**) pressure rise; (**b**) efficiency.

### 2.2. The Computational Mesh and CFD Simulation

The meshing procedure was carried out by dividing the flow domain into four parts, namely, the inlet part, the impeller part (rotating unit), the volute, and the downstream pipe. Structured mesh in ANSYS ICEM-CFD was built to represent the four parts and was converted into unstructured computational meshes. The four mesh parts were then read into ANSYS FLUENT and combined to represent to the whole flow region. Interfaces between neighboring parts were used to interchange flow field data in different parts.

The computational mesh is shown in Figure 2. In the near-wall regions, very fine grid (at least 15 layers) was applied (shown in Figure 2), resulting in many cells. Several grid spaces were considered in successive refinements to check the influence of the grid density on the pressure and efficiency performance of the centrifugal fan, and finally a mesh with 8,975,832 cells was used in the unsteady simulation. The mesh statistics are shown in Table 2, in total, nearly 9 million cells were used. The CFD model without the downstream pipe was used in the study of Lu et al. [25], where LES was carried out. The mesh scheme was found to be not fine enough to performing LES, because according to the recommendation of Versteeg and Malalasekera [26], the non-dimensional wall distance of the first layer in the mesh $y^+ \leq 1$. However, in the simulation the condition, $y^+ \leq 5$ can be satisfied on most wall surfaces, which meets the requirement of the $k-\omega$ turbulent model.

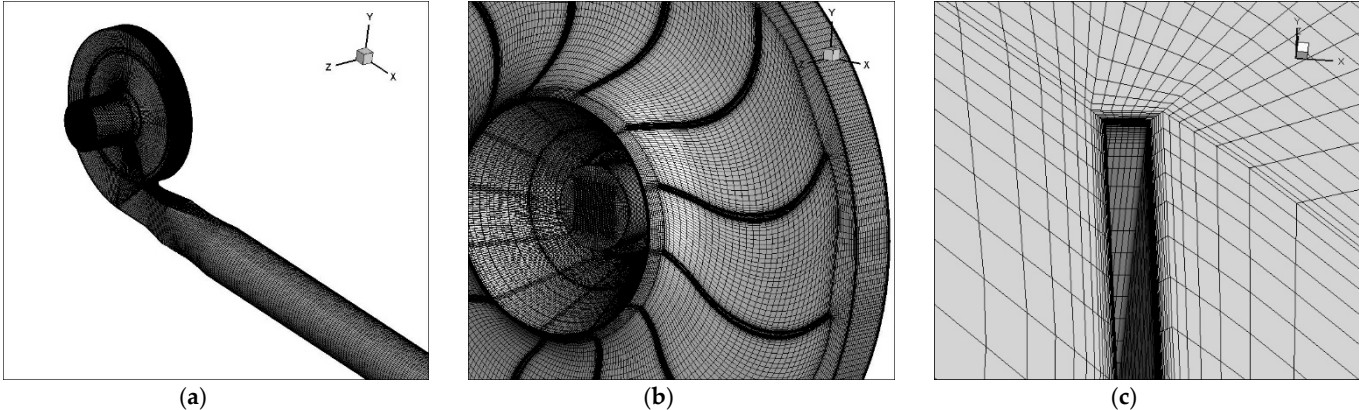

(a)                                             (b)                                             (c)

**Figure 2.** Fan internal flow region mesh: (**a**) the whole model; (**b**) the impeller; (**c**) snapshot of mesh at the blade trailing edge.

**Table 2.** Mesh statistics of the fan.

| Number | Inlet Part | Impeller | Volute | Downstream Pipe |
|---|---|---|---|---|
| Cell | 1,003,684 | 2,119,308 | 2,947,794 | 2,905,046 |
| Node | 974,579 | 2,024,484 | 2,887,526 | 2,869,296 |

The internal flows of the centrifugal fan can be regarded as incompressible because the Mach number with the circumferential velocity at the blade outlet $u_2/c$ = 0.179 (<0.3). Therefore, the incompressible N-S equations were used for computational procedure in this study. With the incompressible assumption, only the hydrodynamic pressure is obtained without the acoustic pressure. The compressible N-S equations can certainly simulate the sound waves as well as turbulence. However, it is quite challenging to numerically resolve the acoustic waves because acoustic intensity is oftentimes five to six orders smaller than the mean flow, and a numerical scheme must have extremely low numerical noise in order to accurately obtain the parameters of acoustic waves [27].

The detached eddy simulation was employed to model the turbulence. The mathematical model can be found in the Appendix A. The numerical simulation was realized using a commercial CFD package ANSYS FLUENT. The impeller domain was set to be rotating with a speed of 2900 RPM to include the centrifugal force source. The remaining three parts, i.e., the inlet part, the volute, and the downstream pipe, were stationary. According to the 0.75BEP, BEP, and 1.49BEP flow rates, the velocity was applied at the inlet with the magnitudes of 9.45, 12.6, and 18.73 m/s, respectively. Medium turbulent level with the turbulent intensity of 5% and the hydraulic diameters of 0.177 were set as the turbulence parameters. The pressure outlet boundary condition with the averaged static pressure of 0 Pa was applied at the pipe outlet surface. The pressure–velocity coupling was calculated using the SIMPLEC algorithm [26]. Second-order, upwind discretizations were used for convection terms, and bounded central differencing schemes were used for diffusion terms. For the time-dependent term scheme, we applied an implicit second-order scheme. The convergence criteria for the residuals of the continuity equation, the momentum equations, and the turbulence equations were set as $1.0 \times 10^{-5}$ during iteration.

A steady-state simulation was initially carried out with the SST $k-\omega$ as the turbulence model, and the results served as the initial conditions for the following unsteady calculation. In the unsteady-state computation, the sliding mesh technique was applied to the interfaces in order to allow unsteady interactions between the impeller and the volute. A complete impeller revolution was divided into 2048 time-steps, i.e., the time step was equal to $1.010237 \times 10^{-5}$ s. In the previous study [28], one impeller revolution was divided into 512 time steps in the URANS simulation with the standard $k-\varepsilon$ equations as the turbulence modelling. Here, for DES, a much finer computational mesh was adopted, and therefore a smaller time step was needed to ensure the stability of the numerical calculation. In the study of [28], only the internal flow in the centrifugal fan was calculated, and with the acoustic analogy theory the aerodynamic noise was predicted, showing that the volute casing noise was predominant. However, in the present study, the unsteady flow in the fan as well as in the long downstream pipe was simulated in order to study the development and transportation of the pressure fluctuations.

The target convergence criteria for the residuals of the governing equations were $1.0 \times 10^{-5}$ during iterations, and a max number of 30 iterations per time step. A representative section of convergence history at BEP flowrate is shown in Figure 3, and good convergence was found. In the unsteady-state simulation, the results of this simulation became statistically stable after several impeller revolutions. Then, the data sampling for time statistics in the ANSYS FLUENT solver was activated and the temporal pressure data of monitoring points were saved at every time step. A total of 16,384 time-steps (8 impeller revolutions) were sampled. The calculation of each flowrate took about 30 days in a cluster with 4 computer nodes. Each node had two Intel Xeon Gold 6132 processors, and one processor has 14 cores; therefore, 112 cores were used during the simulation in total.

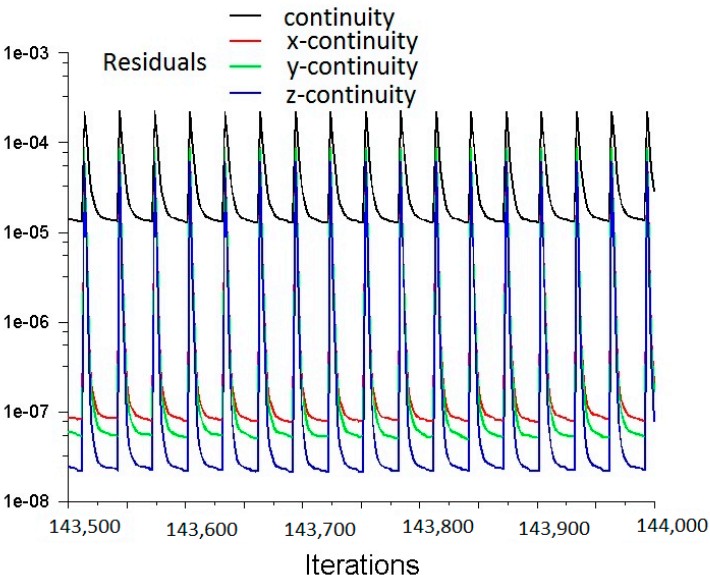

**Figure 3.** The iteration convergence history at BEP in the unsteady simulation.

## 3. Results and Discussion

From the time-step-averaged flow results, we obtained the values of total pressure increase $\Delta p_{tot}$ by the difference of the mass-averaged total pressure at the fan outlet and inlet. The torque of the impeller $T$ was received through the integration of moment in z-direction of the pressure and friction force over the impeller walls. The fan efficiency was calculated on the basis of the equation $\eta = Q\Delta p_{tot}/(T \cdot 2\pi N/60) = 0.767$. The simulation results of fan performance were compared with the experimental results as shown in Figure 1. Good agreement can be found showing the validity of the application of DDES in the centrifugal fan. The numerical results showed slightly better fan performance. It may have due to the fact that other flow loss such as internal leakage had not been resolved well in the flow simulation.

### 3.1. Mean Flow Field

The mean pressure contour on the middle span plane of the volute is shown in Figure 4, where the non-dimensional pressure was obtained by dividing current value of pressure on $p_{\text{ref}}$. During the simulation, the obtained results were saved on different final time steps, which led to different pitch angles of the impeller, but it did not affect the general conclusions of the flow fields with different flowrates. As can be seen from the figure, static pressure rose from the impeller inlet along the impeller and further increased in the downstream pipe where part of the dynamic pressure was converted into static pressure. The pressure value on the blade pressure side was higher than on the suction side. The highest static pressure value was observed around the tongue tip, where the stagnation occurred. According to the velocity triangles in turbomachinery, the outlet flow angle increased with large flow rate. As a result, the stagnation point shifted from the outlet pipe side to the impeller side with the increasing of the flowrate. The impeller side of the volute tongue suffered a low-pressure region, where the air was sucked back into the volute with the increase of the velocity.

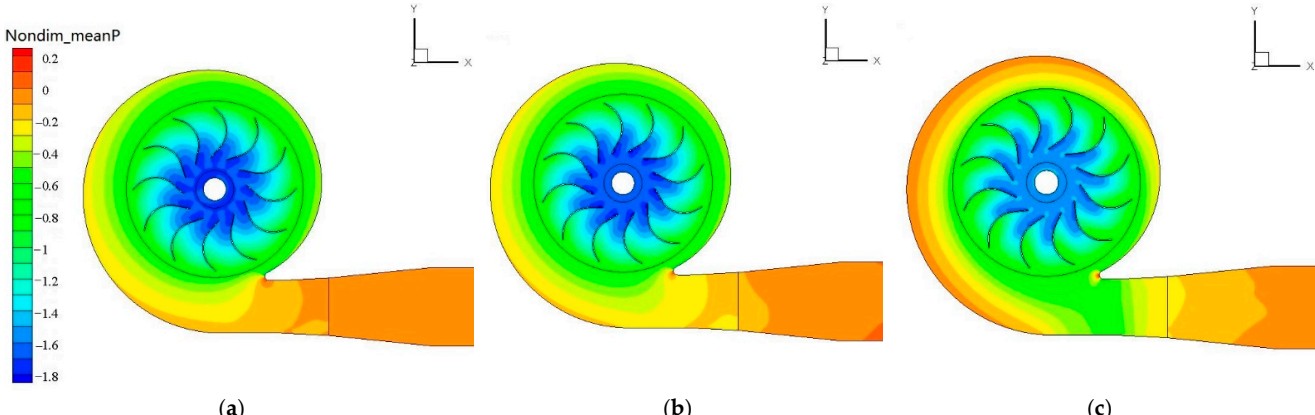

**Figure 4.** The mean pressure contour at the middle span of the volute: (**a**) 0.75BEP; (**b**) BEP; (**c**) 1.49BEP.

The averaged velocity magnitude distribution on the middle span of the volute is shown in Figure 5. As we can see from the figure in the tongue region, the flow was split at the tongue tip, resulting in the stagnation point therein at the nominal flow rate. The maximum absolute velocity was located at the blade trailing edge regions, which can reach 1.5 times of the circumferential velocity at the blade trailing edge $u_2 = 60.73$ m/s. Forward-curved blade can lead to large absolute velocity value at the impeller outlet according to the velocity triangle of a rotor in turbomachinery theory.

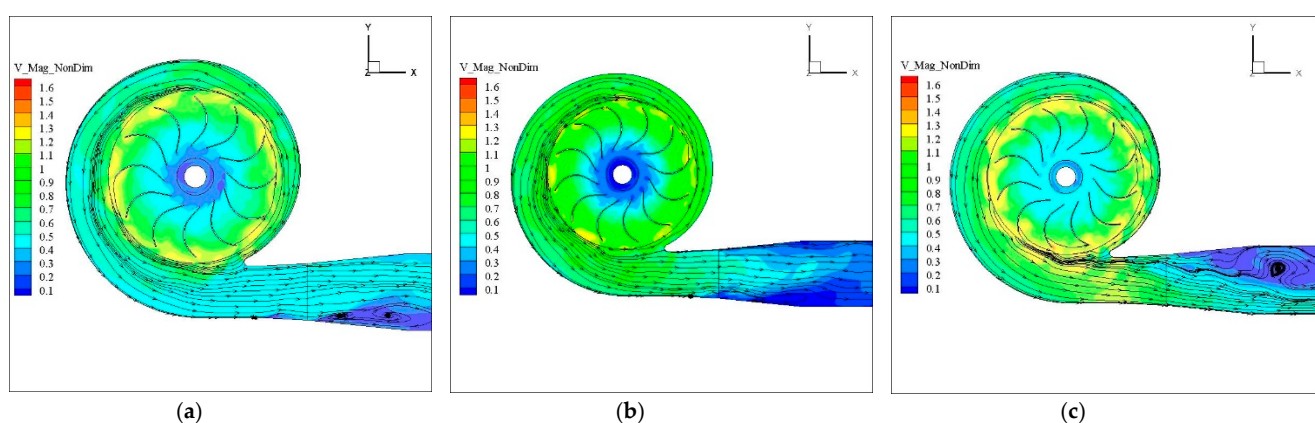

**Figure 5.** The mean velocity distribution at the middle span of the volute: (**a**) 0.75BEP; (**b**) BEP; (**c**) 1.49BEP.

At the impeller outlet, the velocity on the blade pressure side was larger than its value on the suction side. The circumferential non-uniform patterns of the velocity and pressure at the impeller outlet generated the regular pressure fluctuations on the volute casing surface, especially on the volute tongue surface that was close to the impeller. When a blade passed by, a pulsation occurred, and thus the pressure fluctuations had the BPF as the fundamental frequency.

At 0.75BEP flowrate, the flow leaving the impeller had a small flow angle. As a result, a vortex region existed at the lower part of the transitional pipe, as can be seen in Figure 5a. At 1.49BEP flowrate, the flow angle increased, and the flow tended to attack the lower part of the outlet pipe. Thus, a vortex region formed at the upper part of the transitional pipe (see Figure 5c).

The turbulent kinetic energy $k$, the turbulent specific dissipation $\omega$, and the vorticity were similar for the three flowrates. Therefore, only the 1.49BEP case is shown in Figure 6, which was taken as the example for explanation. Large turbulent kinetic energy exited in the places of the volute tongue region and the impeller outlet. The lower part of the transitional section of the downstream pipe also suffered intensive turbulence. Checking

the streamlines in Figure 5c we found that vortices formed in that region due to the divergent shape of the transitional pipe.

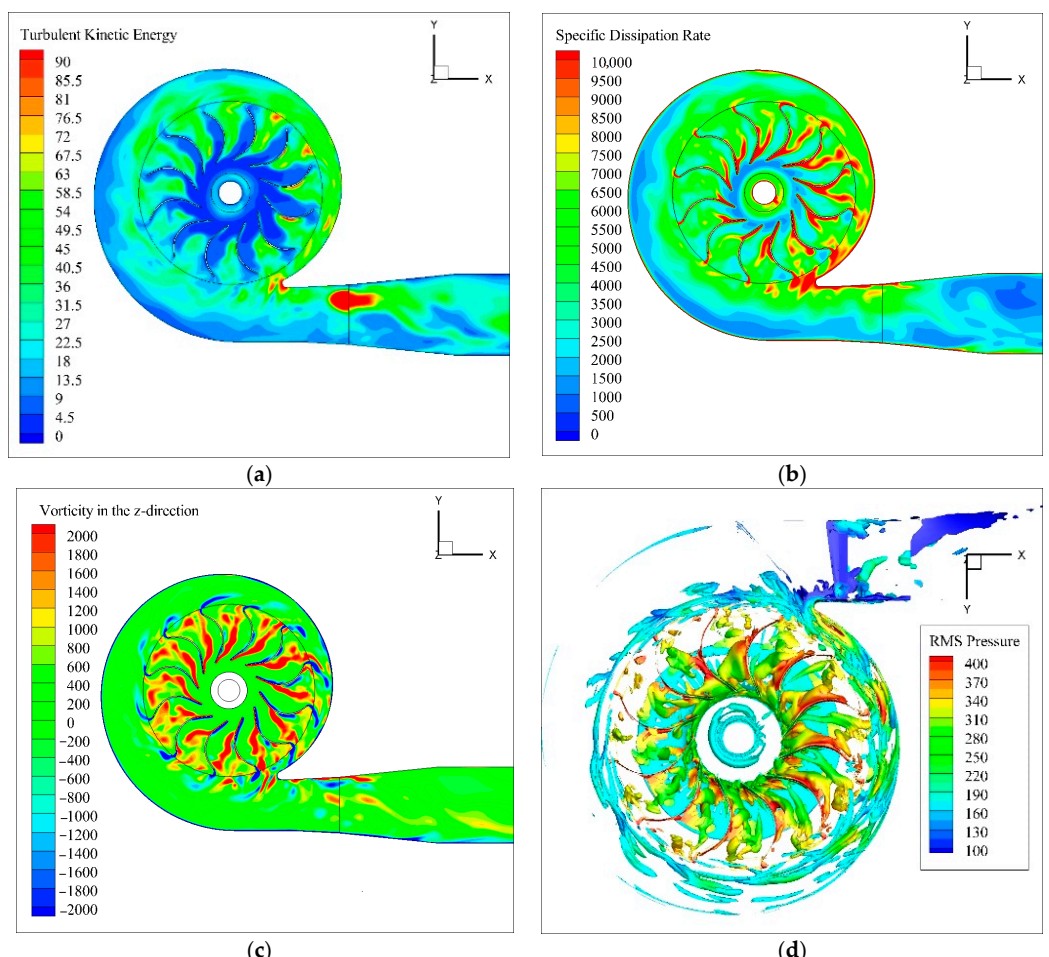

**Figure 6.** (**a**) Turbulent kinetic energy $k$ (m$^2$/s$^2$), (**b**) turbulent specific dissipation $\omega$ (1/s), (**c**) vorticity in the z-direction at the middle span of the volute, and (**d**) the vortices identified by Q-criterion ($Q^* = 0.01$) at 1.49BEP flowrate.

As we see from Figure 6b large turbulent specific dissipation values $\omega$ were observed near wall regions where large velocity gradients existed. In the blade channel, the turbulent-specific dissipation was also quite large. The reason can be found by analyzing z-direction vorticity $\omega_z$ in Figure 6c Large negative values of $\omega_z$ were observed around the blade trailing edges, on the pressure side of the beginning part of blades, suction side of the blade middle part, while large values of positive value $\omega_z$ existed in the blade channels. In large vorticity regions, fluid particles had a large velocity difference resulting in large turbulent energy dissipation.

The 3D vortical structure can be examined using the Q-criterion, which is represented as

$$Q = \frac{1}{2}\left[|\Omega|_2 - |S|^2\right] \tag{1}$$

where $\Omega$ and $S$ are the second invariant measure of the vorticity and strain rate tensors

$$\mathbf{\Omega} = \frac{1}{2}\left[\nabla\mathbf{u} - (\nabla\mathbf{u})^T\right], \mathbf{S} = \frac{1}{2}\left[\nabla\mathbf{u} + (\nabla\mathbf{u})^T\right] \tag{2}$$

A positive $Q$ value indicates that the rotating effect is stronger than the strain rate effect. The non-dimensional value of $Q$ is calculated by [24]

$$Q^* = \sqrt{Q/Q_{max}} \tag{3}$$

where $Q_{max}$ is the maximum $Q$ value. Figure 6d shows the iso-surface of $Q^* = 0.01$ with the $p_{rms}$ as the contour color. In order to see the vortices more clearly, we took the view angle from the hub side because of the vortex sheet on the shroud side. Here, we can see the large vortical structures on the blade suction sides and in most blade channels. Vortices also appeared around the impeller wheel exit, where airflow discharged into the volute, causing expansion. At the beginning of the spiral volute, intensive vortices existed, especially in the tongue region, where interactions between the rotating impeller and the stationary volute was the strongest.

The distributions of the pressure and velocity along the central line of the downstream pipe are shown in Figure 7, in which the distance refers to the distance from the inlet section of the transitional pipe. The downstream pipe had an inner diameter of $D = 190$ mm, and its wall thickness was 5 mm; therefore, the outer radius was $D = 200$ mm. As one can see, the mean pressure increased through the transitional pipe, owing to its diffuser shape, and then decreased gradually due to the flow loss along the pipe. The velocity decreased through the transitional pipe part and reached a constant value where the pipe had a uniform circular section with a diameter of 190 mm.

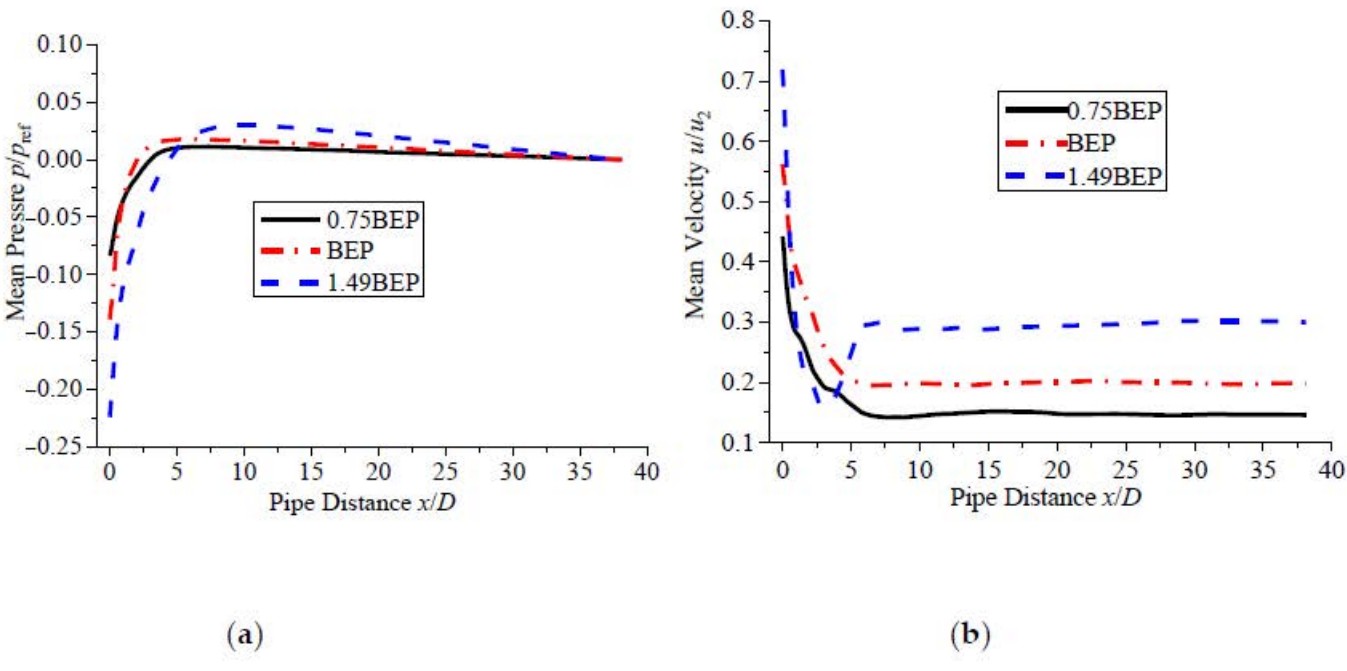

**Figure 7.** (**a**) Pressure and (**b**) velocity characteristics along the downstream pipe.

### 3.2. Fluctuating Flow Field

The root-mean-square (RMS) value was taken as the indicator of fluctuating strength. The RMS value of time history data of any quantity $\varphi$ is defined as follows:

$$\varphi_{RMS} = \sqrt{\sum_{i=1}^{n}(\varphi_i - \overline{\varphi})^2/n} \tag{4}$$

where $\overline{\varphi}$ means the average value. Note that with this equation, the RMS value indicates only the fluctuations, since the mean value is subtracted. The volume-averaged RMS pressure of the four computational domains is shown in Figure 8. The pressure fluctuation

was the smallest at BEP. At 0.75BEP, flow rate pressure fluctuations increased slightly, while at 1.49BEP, flow rate the pressure fluctuations increased noticeably, especially in the impeller domain. In the downstream pipe, the pressure fluctuation was much smaller in comparison with the other parts, which denotes that hydrodynamic pressure fluctuations decreased quickly in the downstream pipe.

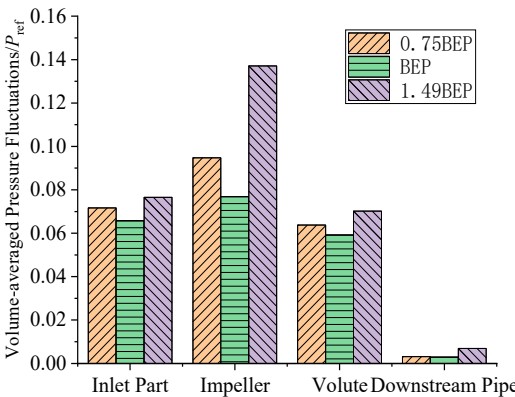

**Figure 8.** The volume-averaged RMS pressure in the computational domains.

The RMS pressure distribution on the middle span of the volute is shown in Figure 9 Inside the impeller, the RMS pressure was around 10–20% of the reference dynamic pressure $p_{ref}$. The pressure fluctuation was more intensive on the blade suction side in comparison with the pressure side for forward curved blades. In this case, flow separation can occur more easily on the blade suction side, especially in the large flowrate condition when the flow attacks the blade pressure side. Flow separation can lead to flow fluctuations.

The maximum pressure fluctuations occurred at the volute tongue, wherein the strongest interaction and unsteadiness were induced by the impeller rotation relative to the fixed volute. The RMS pressure value was able to reach above 30% of $p_{ref}$ at 1.49BEP flowrate. The location of the maximum pressure fluctuations was not at the stagnation point but at the impeller side of the tongue where the air flow was suctioned back to the volute. In consequence, the tongue region was the main aerodynamic sound source of centrifugal fans. Modifications of the volute tongue can reduce the fan noise effectively, for example, through the usage of inclined-tongue [29] and step tongue [30].

The volume-averaged RMS value distribution of velocity is shown in Figure 10 with logarithmic scale usage. The velocity in the impeller was the relative velocity observed from the rotating impeller, and its fluctuation was quite large in comparison with the absolute velocity fluctuations in the stationary parts. As we can see from the figure inside the centrifugal fan as well as the downstream pipe, the intensity of velocity fluctuation increased with the flowrate. The velocity fluctuation was sufficiently small in the inlet part. It is explained by the fact that the airflow suffered fluctuations when it went through the rotating impeller, and the fluctuations were carried to downstream with little influence on the inlet part at the upstream. In the inlet part, the velocity fluctuation was the smallest at BEP because the airflow angle was consistent with the blade inlet angle, causing the weakest incidence.

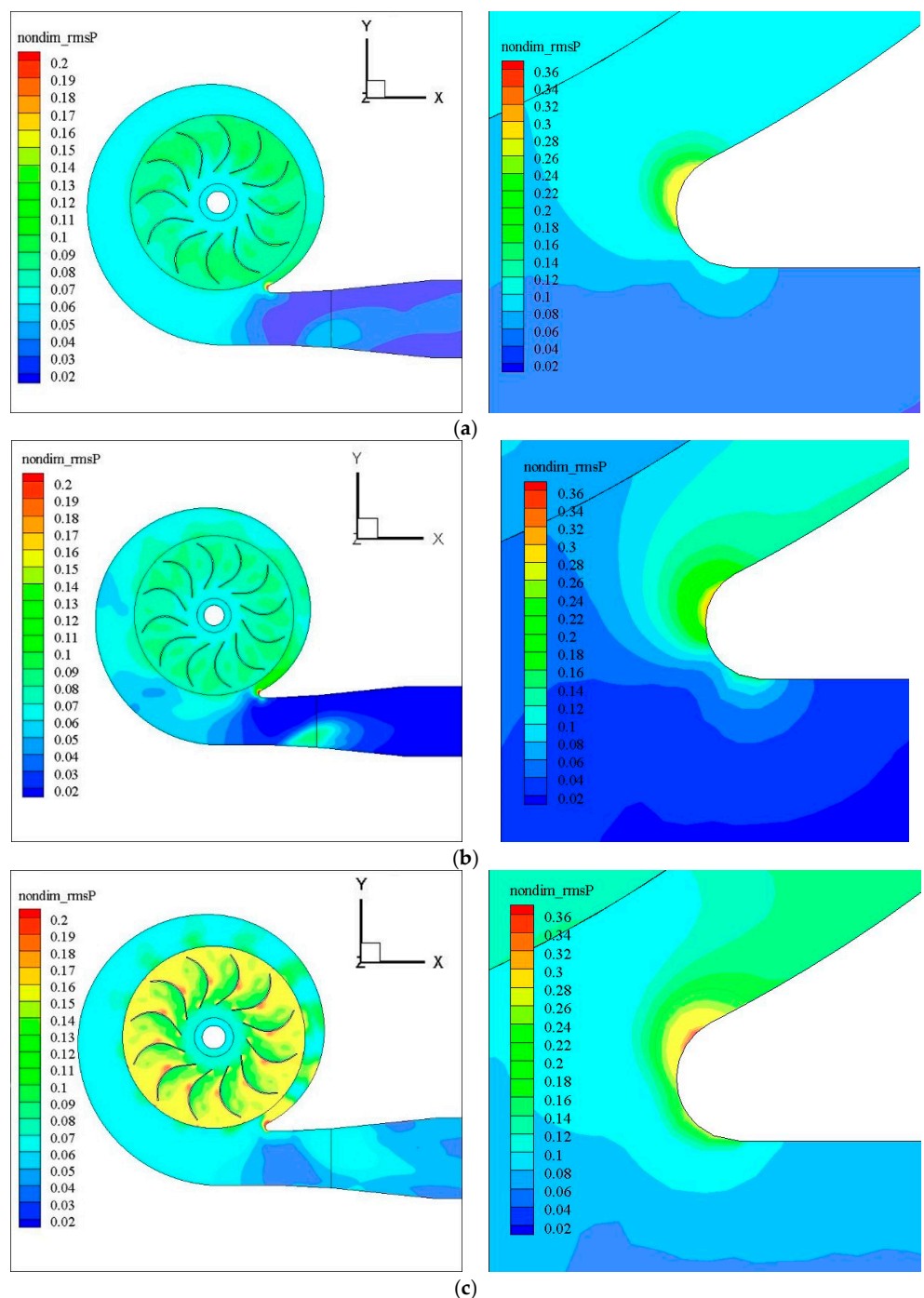

**Figure 9.** The RMS pressure contour at the middle span of the volute: (**a**) 0.75BEP; (**b**) BEP; (**c**) 1.49BEP.

The RMS value distribution of velocity magnitude on the middle span of the volute is shown in Figure 11. As can be seen, the impeller suffered great velocity fluctuations during the rotation, and its magnitude was approximately in the same order with $u_2$. Unlike the pressure fluctuation, which had the minimum value at BEP, velocity fluctuations increased with the flowrate. Intensive fluctuations were found at the impeller outlet and in the volute tongue region, as was expected. The magnitude was around 20% of $u_2$ for BEP and 0.75BEP flowrates; however, for 1.49BEP flowrate, it could be over 30% of $u_2$ in the volute tongue region. For low flow rates, large velocity fluctuation was detected at the lower part of the transitional pipe, and according to Figure 5a,b, we can see that the flow separates in that region due to the diffuser configuration, hence vortices appeared. For large flow rate, large velocity fluctuation exited at the upper part of the transitional pipe because the flow

leaving the impeller had a larger flow angle and tended to attack the low part of the outlet pipe, causing a wake region at the upper part of the transitional pipe, as explained in Section 3.1.

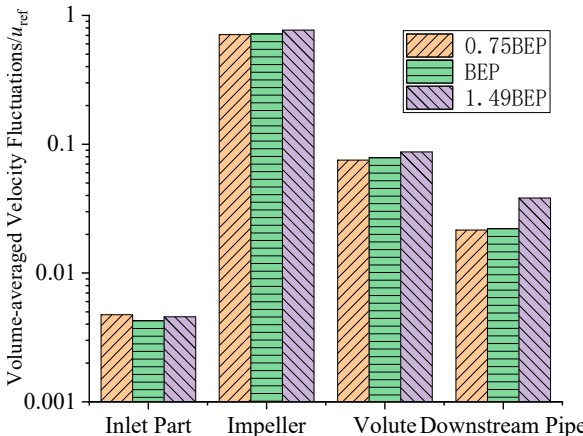

**Figure 10.** The volume-averaged RMS velocity in the computational domains.

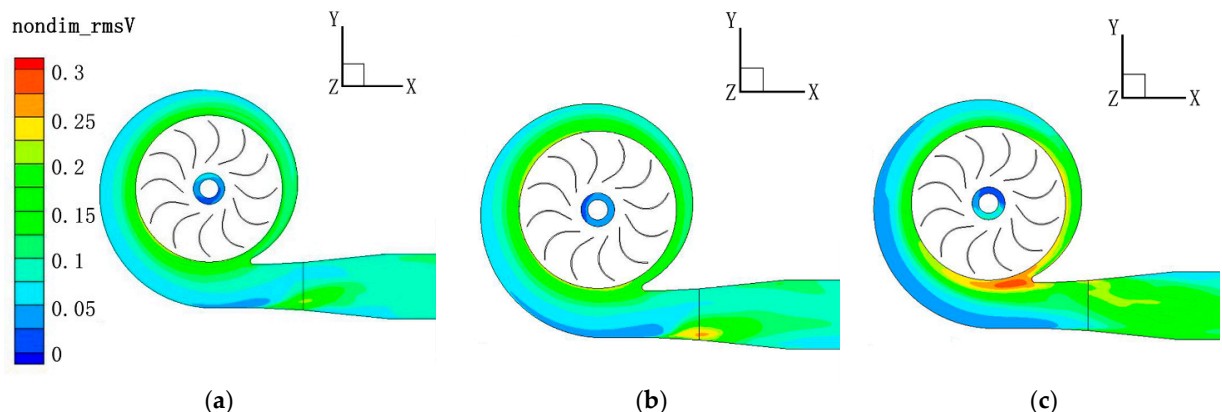

**Figure 11.** The RMS velocity distribution at the middle span of the volute: (**a**) 0.75BEP; (**b**) BEP; (**c**) 1.49BEP.

After the flow entered the downstream pipe, it became smoother in a way that the pressure fluctuations decreased sharply, as can be seen in Figure 12, where the RMS pressure and velocity values along the pipe centerline are shown. The pressure fluctuation intensity may have increased slightly at the inlet of the transitional pipe, which can be ascribed to the enhancement of the local flow disturbance due to the divergent shape, as shown in Figure 5. Then, the pressure fluctuations decreased sharply after the transitional section of the downstream pipe. For the 0.75BEP and BEP flowrates, the pressure fluctuation decreased to 0.5% $p_{\text{ref}}$ after 5D, and about 1% $p_{\text{ref}}$ for the 1.49BEP flowrate. The decaying trend of the velocity fluctuations demonstrated a similar pattern with that of the pressure fluctuations, except that in the transitional pipe section, the velocity fluctuation decreased slower than the pressure fluctuations. This means that local changes of flow passages had more influences on velocity fields than pressure fields.

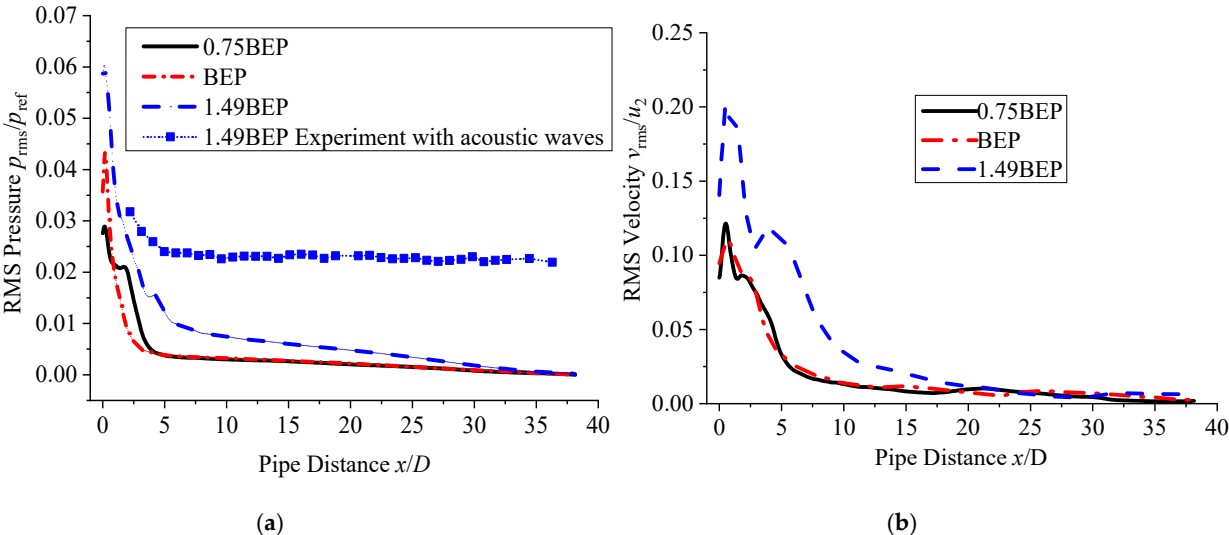

**Figure 12.** The (**a**) RMS pressure and (**b**) RMS velocity along the pipe center line.

Experiment measurements of the pressure fluctuations in the downstream pipe were carried out. The downstream pipe was made of PVC material. It was drilled easily with small holes of 3 mm diameter along the axial direction. The Kulite miniature pressure transducer with the model XTL-140M was used to measure the dynamic pressure. Its tap diameter was 2.54 mm, and it was inserted into the pipe with the depth of 12.5 mm. The analog voltage signal was acquired by Simcenter SCADAS XS with the 24-bit A/D converter. The measurement was repeated eight times, and the standard deviation of the results was found to within 5%.

The measured pressure fluctuation at the 1.49BEP flowrate is also shown in Figure 12a. Here, it should be pointed out that the numerical results were along the centerline while the experimental results were received along the line, which was 12.5 mm away from the wall surface. We can see from the figure that numerical pressure fluctuation had the same order of magnitude in comparison with the numerical result. Moreover, the trend was also similar. However, the experimental data showed higher pressure fluctuation intensity through the whole downstream pipe. This may be ascribed to the acoustic signals in the experimental data, which were completely ignored in numerical simulations with incompressible flow models.

From the study, we can conclude that the hydrodynamic fluctuation faded quickly as soon as the flow left the centrifugal fan, and acoustic fluctuations were dominant thereafter.

### 3.3. Spectra of Pressure Fluctuations

Several monitoring points were set inside the flow domain to record the pressure time history data (see Figure 13a). Vlt1~Vlt7 were seven points on the casing wall along the volute profile at the mid-span of the volute: Vlt1 was taken at the tongue tip, with every 60° in the circumferential direction setting for Vlt2~Vlt6, and Vlt7 was 30° away from Vlt6. Pp00~Pp10 were taken along the centerline of the downstream pipe: Pp00 was at the fan outlet, Pp0 was at the transitional pipe outlet, Pp1 was 0.2 m from the Pp0, and Pp5 and Pp10 were respectively 1 and 2 m from Pp0.

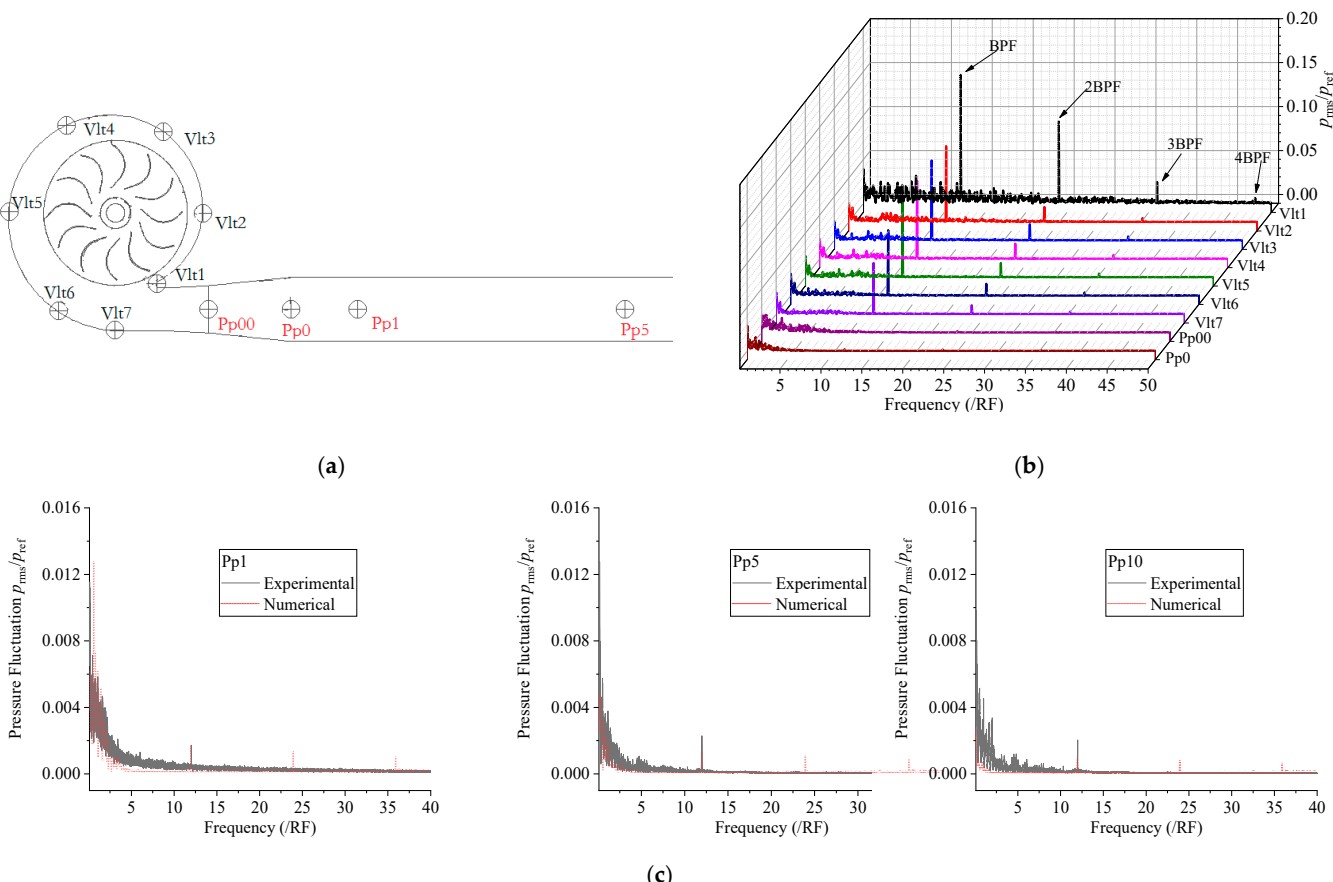

(a)

(b)

(c)

**Figure 13.** (**a**) The monitoring points and (**b**,**c**) the spectra of pressure fluctuations of 1.49BEP flowrate.

The 1.49BEP flowrate was selected as an example to show the spectra of the pressure fluctuations of monitoring points (see Figure 13b,c). For the pressure fluctuations in the volute, the discrete components of BPF and its second and third harmonics were noticeable in the spectra. The pressure fluctuation amplitudes of the monitoring points in the volute were larger than those in the downstream pipe, and the volute tongue had the most intensive flow fluctuations, as we already expected. At the volute tongue, the BPF component was able to reach a value of 330 Pa, which was nearly 15% of $p_{ref}$, and this component decreased quickly, which can be described in mathematical way according to the elliptical Poisson equation of pressure (see Equation (A3) in the Appendix A). After 5$D$ distance in the downstream pipe ($D$ is the pipe diameter), the BPF component reduced to 2.9 Pa, i.e., less than 0.2% of $p_{ref}$. At the beginning of the downstream pipe, low-frequency hydrodynamic pressure fluctuations were noticeable, but after distance was equal to 10$D$, the pressure fluctuations decreased to being lower than 0.1% of $p_{ref}$, similar to the BPF component.

From the experimental results shown in Figure 13c, we can see that at the beginning of the downstream pipe, they agreed with each other to a certain extent, showing that hydrodynamic pressure fluctuations were dominant. The hydrodynamic pressure fluctuations, especially the low-frequency components, reduced quickly along the pipe. However, the pressure fluctuations decreased much more slowly than the hydrodynamic pressure fluctuations because most fluctuations were related to acoustic pressures. From the measured pressure spectra, we can see that at Pp5 and Pp10, the spectra of experimental results were similar; thus, the overall intensity of pressure fluctuations was almost the same at these two points, as shown in Figure 11a.

## 4. Conclusions

The unsteady hydrodynamic flow fields of a centrifugal fan with a long downstream pipe, at BEP, 0.75BEP, and 1.49BEP flowrates, were analyzed in detail using CFD with the delayed detached eddy simulation (DDES) as the turbulence modeling. The turbulence and vorticity characteristics are discussed. The flowfields revealed the characteristics of impeller–volute interactions, which generated the pressure and velocity fluctuations with the blade-passing frequency (BPF) as the predominant component.

The time-averaged flow fields clearly showed that the pressure increased gradually through the impeller, further recovered from the velocity in the volute, and gradually decreased along the downstream pipe due to friction. Owing to the forward-curved blades, at the impellor exit, the flow had very large velocity, and the maximum velocity near the trailing edge was able to reach 1.5 $u_2$, where $u_2$ is the blade velocity at the impeller outlet. The root-mean-square value pressure distributions showed that most places inside the centrifugal fan underwent large pressure fluctuations with a magnitude of about 10% of the reference dynamic pressure $p_{ref} = 0.5 \rho u_2^2$; the maximum value located at the tongue tip was able to reach 30% of $p_{ref}$. Unlike the pressure fluctuation, which had a minimum value at BEP, velocity fluctuations increased with the flowrate. The velocity fluctuation was intensive at the impeller outlet, and its magnitude was around 20% of $u_2$ in the volute tongue region, but for 1.49BEP flowrate, it can be over 30% of $u_2$ in the tongue region.

The pressure fluctuation magnitude decreased rapidly along the outlet pipe: after 5$D$ ($D$ is the outlet pipe diameter), the magnitude was about 0.5% $p_{ref}$ for the 0.75BEP and BEP flowrates, and 1% $p_{ref}$ for the 1.49BEP flowrate. The discrete components at BPF and its second and third harmonics were prominent in the pressure fluctuation spectra at the monitoring points in the volute. The BPF component had a maximum value of 15% of $p_{ref}$ in the tongue region at the 1.49BEP flowrate, and it decreased sharply along the downstream pipe with an amplitude less than 0.2% of $p_{ref}$ after 5$D$ distance.

The spectra of the hydrodynamic pressure were compared with the measured pressure fluctuations at 1.49BEP flowrate. At the beginning of the downstream pipe, they agreed with each other to a certain extent, showing that hydrodynamic pressure fluctuations are dominant wherein, however, the experimental pressure fluctuations decreased more slowly than the hydrodynamic pressure due to the acoustic pressure waves in the downstream pipe.

This study focused on the internal flow of the centrifugal fan as well as its downstream pipe, and it is expected to contribute to the deep understanding of the hydrodynamic unsteady flow. Future study will include acoustic fluctuations that will be dominant in places away from disturbance such as in the downstream pipe.

**Author Contributions:** J.-C.C.: writing—original draft, writing—review and editing, funding acquisition, investigation, methodology; H.-J.C.: investigation; V.B.: writing—original draft, investigation. Y.-H.G.: supervision, writing—review and editing. All authors have read and agreed to the published version of the manuscript.

**Funding:** This research was funded by the National Natural Science Foundation of China (grant number 51976201) and Zhejiang Provincial Natural Science Foundation of China (grant number LY18E060006).

**Institutional Review Board Statement:** Not applicable.

**Informed Consent Statement:** Not applicable.

**Data Availability Statement:** The data that supports the findings of this study are available within the article.

**Acknowledgments:** The authors kindly acknowledge the support of Nanfang Ventilator Co., Ltd. (Foshan, Guangdong Province, China).

**Conflicts of Interest:** The authors declare no conflict of interest.

**Appendix A. Mathematical Model and Numerical Method**

*Appendix A.1. Governing Equations*

For incompressible flow, the continuity equation and the momentum equation read

$$\nabla \cdot \mathbf{u} = 0 \tag{A1}$$

$$\frac{\partial \mathbf{u}}{\partial t} + (\mathbf{u} \cdot \nabla)\mathbf{u} = -\frac{1}{\rho}\nabla \overline{p} + \nu\nabla^2\mathbf{u} \tag{A2}$$

where $\mathbf{u}$ is the flow velocity vector; $p$ is the pressure; and $\rho$ and $\nu$ are the density and kinematic viscosity of the fluid, respectively. The Poisson equation of the pressure can be obtained by performing the divergence of the momentum Equation (A2) and considering the divergence free velocity condition (A1), leading to

$$\nabla^2 p = -\rho\nabla \cdot (\mathbf{u} \cdot \nabla)\mathbf{u} \tag{A3}$$

The Poisson equation of pressure is elliptic with the velocity field of the sources, and the pressure is referred to as the hydrodynamic pressure, which decays quickly away from their source of generation. Unlike the hydrodynamic pressure fluctuation governed by the Poisson Equation (A3), the acoustic equation is

$$\frac{1}{c^2}\frac{\partial^2 p'}{\partial t^2} - \nabla^2 p' = 0 \tag{A4}$$

where $c$ is the speed of sound, and $p'$ is the acoustic pressure. This is the partial differential equation of hyperbolic type, and the acoustic pressure fluctuations pass through the pipe without attenuation if damping is not taken into account. For low Mach number flow, the famous acoustic analogy theory by Lighthill [31] reformulates the continuity equation and momentum equation into a nonhomogeneous wave equation:

$$\frac{\partial^2 \rho'}{\partial t^2} - c_\infty^2\frac{\partial^2 \rho'}{\partial x_i^2} = \frac{\partial^2 T_{ij}}{\partial x_i \partial x_j} \tag{A5}$$

With the Lighthill's tensor,

$$T_{ij} = \rho v_i v_j + (p - p_\infty) - (\rho - \rho_\infty)c_\infty^2\delta_{ij} - \sigma_{ij} \tag{A6}$$

where the acoustic density variation is $\rho' = c_\infty^2 p'$, the sub $_\infty$ means the reference value of ambient condition, $\delta_{ij}$ is the Kronecker delta, and $\sigma_{ij}$ is viscous stress tensor. Thus, aerodynamic noise generated by the unsteady flow in the centrifugal fan can propagate through the downstream pipe as the acoustic waves causing acoustic pressure fluctuations.

Computational aeroacoustics is not the same as CFD, and it faces a different set of computational challenges, as were pointed out by Tam [27]. In the present study, the acoustic pressure was ruled out due to the incompressible flow model.

*Appendix A.2. Turbulence Model*

It can be concluded that the internal flow is turbulent in nature on the basis of the characteristic Reynolds number, Re $= du_2/\nu = 1.57 \times 10^6$. In turbulence modelling, the flow is usually separated into the mean and fluctuating components as $\phi = \overline{\phi} + \phi'$. With this decomposition, the governing N-S equations read as follows:

$$\nabla \cdot \overline{\mathbf{u}} = 0 \tag{A7}$$

$$\frac{\partial \overline{\mathbf{u}}}{\partial t} + \nabla \cdot (\overline{\mathbf{u}}\,\overline{\mathbf{u}}) = -\frac{1}{\rho}\nabla \overline{p} + \nu\nabla^2\overline{\mathbf{u}} - \frac{1}{\rho}\nabla \cdot \boldsymbol{\tau}^R \tag{A8}$$

where $\boldsymbol{\tau}^R$ is the turbulent or Reynolds stress tensor $\boldsymbol{\tau}^R = \rho\overline{\mathbf{u}'\mathbf{u}'}$. The overbar should be interpreted as the ensemble average procedure in RANS approach and as the spatial filtering operation in LES. The N-S equations are closed by boundary conditions: either a Dirichlet condition $\overline{\mathbf{u}}$ or a Neumann condition $\boldsymbol{\sigma} \cdot \mathbf{n} = \overline{\mathbf{t}}$ is specified, where $\boldsymbol{\sigma}$ and $\mathbf{n}$ are the stress tensor and unit outward normal vector, respectively, and $\overline{\mathbf{t}}$ is the specified tensional force. With the help of the constitutive model of the fluid, the directional derivatives of $\overline{\mathbf{u}}$ are specified in this manner.

An improved version of detached eddy simulation, namely, delayed detached eddy simulation (DDES), was used in this study to calculate the turbulence. In DES, the Reynolds stress is as follows by the Boussinesq approximation:

$$\boldsymbol{\tau}^R = \rho\overline{\mathbf{u}'\mathbf{u}'} = 2\mu_T\overline{\mathbf{S}} - \frac{2}{3}\rho k\mathbf{I} \tag{A9}$$

where $\overline{\mathbf{S}}$ is the mean strain rate tensor, $\mu_T$ is the turbulent eddy viscosity, $k$ is the turbulent kinetic energy, and $\mathbf{I}$ is the identity tensor.

In the context of RANS simulations, the SST $k-\omega$ turbulence model [26] was chosen for its advantages in simulating the near-wall flows. The SST $k-\omega$ turbulence mode employs the $k-\omega$ model in the boundary layer flows and switches to the $k-\varepsilon$ model in free shear streams. For the $k-\omega$ model, the turbulent kinetic energy $k$ and turbulent-specific dissipation $\omega$ are descripted by the transport equations [22]:

$$\frac{\partial k}{\partial t} + \nabla \cdot (\overline{\mathbf{u}}k) = P_k \\ + \nabla \cdot [(\nu + \nu_T/\sigma_k)\nabla k] - \sqrt{k^3}/l_{DDES} \tag{A10}$$

$$\frac{\partial \omega}{\partial t} + \nabla \cdot (\overline{\mathbf{u}}\omega) = \nabla \cdot [(\nu + \nu_T/\sigma_\omega)\nabla\omega] + \alpha P_k/\nu_T \\ - \beta\omega^2 + 2(1 - F_1)\sigma_{\omega 2}(\nabla k \cdot \nabla\omega)/\omega \tag{A11}$$

where $\alpha, \beta, \sigma_k, \sigma_\omega, \sigma_{\omega 2}$ are model constants, the kinematic eddy viscosity $\nu_T = \mu_T/\rho$. The $F_1$ is the first blending function for an automatic switching between the $k-\omega$ and $k-\varepsilon$ models, and $P_k$ is the function of turbulent kinetic energy production:

$$P_k = \min(\boldsymbol{\tau}^R : \nabla\overline{\mathbf{u}}, 10\beta k\omega). \tag{A12}$$

The turbulent eddy viscosity $\mu_T$ is bounded by

$$\mu_T = \frac{\rho a_1 k}{\max(a_1\omega, SF_2)}, \tag{A13}$$

where $a_1 = \sqrt{\beta}$, $S = \sqrt{2(\overline{S}_{ij}\overline{S}_{ij})}$ is the second invariant of the strain rate tensor, and $F_2$ is a second blending function with the expression as

$$F_2 = \tanh\left[\left[\max\left(\frac{2\sqrt{k}}{\beta^*\omega y}, \frac{500v}{y^2\omega}\right)\right]^2\right]. \tag{A14}$$

The DDES turbulent length scale in Equation (A10) can be written as follows:

$$l_{DDES} = l_{RANS} - f_d\max(0, l_{RANS} - l_{LES}) \tag{A15}$$

where $l_{LES} = C_{DES}h_{\max}$, $l_{RANS} = \sqrt{k}/(C_\mu\omega)$, $h_{\max}$ is the maximum edge length of the cell., and

$$C_{DES} = C_{DES1}F_1 + C_{DES2}(1 - F_2) \tag{A16}$$

The model constants of the above equations can be found in [22].

*Appendix A.3. Numerical Method*

The finite volume method is employed to discretize the integral form of the mass and momentum equations or every control volume $V_\mathrm{P}$ at point P as follows:

$$\sum_f \overline{\mathbf{u}}_f \cdot \mathbf{n}_f S_f = 0 \tag{A17}$$

$$\frac{\partial}{\partial t} \overline{\mathbf{u}}_P V_\mathrm{P} + \sum_f \left( \overline{\mathbf{u}}_f - \mathbf{u}_m \right) \cdot \mathbf{n}_f \overline{\mathbf{u}}_f S_f = \sum_f \nu_f \mathbf{n}_f \cdot (\nabla \overline{\mathbf{u}})_f S_f - \frac{1}{\rho} (\nabla p)_\mathrm{P} V_\mathrm{P} \tag{A18}$$

where the subscript P represents the volume values, and f means the face values. The $V_\mathrm{P}$ is the cell volume of each control volume, $S_f$ is the face area, and $\mathbf{n}_f$ is the unit outward normal vector of the face. The transport equations of the turbulent kinetic energy $k$ and turbulent specific dissipation $\omega$ are discretized in the same manner as in Equation (A18) when RANS simulation is performed instead of LES. At the centroid of each cell, the variable values are calculated directly, and for variable values at the control volume surface, suitable interpolation is used with the center values. Finally, suitable quadrature formulae are applied to approximate the surface and volume integrals to a second-order accuracy in this study.

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
