# Peer review of "Study of the Hydrodynamic Unsteady Flow Inside a Centrifugal Fan and Its Downstream Pipe Using Detached Eddy Simulation"

_sustainability, doi:10.3390/su13095113_

Round 1

Reviewer 1 Report

Please see the attached report for comments.

In the current manuscript, authors investigate unsteady hydrodynamic flow fields of a centrifugal fan using Ansys and Delayed Detached Eddy Simulation turbulence modeling. Two key strengths of the paper is a good introduction section and a good analysis and discussion of the results. Authors also carried out grid-independence study to make sure the results are independent of the grid size.
The manuscript can be recommended for the publication once the following comments are addressed:
1. On line 287, a previous study [28] is mentioned, but no details regarding how that study is related to the current study are provided. Authors should summarize the key findings of Ref. [28] and discuss how that study is relevant.
2. Several references are missing and have “Error! Reference source not found” message instead. This should be fixed and it is always a good practice to carefully proofread the manuscript before submitting it, in order to avoid such mistakes.

Author Response

Dear anonymous reviewer,

We would like to appreciate your review comments. We have carefully revised the manuscript according to your suggestion. Please find our responses to the comments in the following.

Reviewer 1

In the current manuscript, authors investigate unsteady hydrodynamic flow fields of a centrifugal fan using Ansys and Delayed Detached Eddy Simulation turbulence modeling. Two key strengths of the paper is a good introduction section and a good analysis and discussion of the results. Authors also carried out grid-independence study to make sure the results are independent of the grid size.
The manuscript can be recommended for the publication once the following comments are addressed:
C1. On line 287, a previous study [28] is mentioned, but no details regarding how that study is related to the current study are provided. Authors should summarize the key findings of Ref. [28] and discuss how that study is relevant.

A1: Thank you for your suggestion. In the revised manuscript, we have added the key findings of Ref. [28] and its relation as well as the differences with the present study as follows: “In the previous study[28], one impeller revolution was divided into 512 times steps in the URANS simulation with the standard k-ε equations as the turbulence modelling. Here for DES, a much finer computational mesh is adopted, therefore a smaller time step is needed to ensure the stability of the numerical calculation. In that study [28], only the internal flow in the centrifugal fan was calculated, and with the acoustic analogy theory the aerodynamic noise was predicted showing that the volute casing noise is the predominant. While in the present study the unsteady flow in the fan as well as in the long downstream pipe is simulated in order to study the development and transportation of the pressure fluctuations.”

C2. Several references are missing and have “Error! Reference source not found” message instead. This should be fixed and it is always a good practice to carefully proofread the manuscript before submitting it, in order to avoid such mistakes.

A2: Thank you for your reminder. We double checked the reference citations in the revised manuscript. Actually, we use Endnote to cite the references and Mathtype to write equations and insert equation references. It may cause some errors when the software is not installed.

Reviewer 2 Report

This paper presents a numerical investigation of the unsteady flow field within a centrifugal fan, containing a long downstream pipe, at three different inlet flow rates, via detached eddy simulations. Both time-averaged and time-resolved flow characteristics are evaluated, serving as a tool to understand the hydrodynamic unsteady regime.

The authors should implement the following revisions before the paper is accepted to publication:

  • The authors should move section 2.2 to appendix

  • In Figure 2, the authors should include a snapshot of near-wall grid

  • The authors should quantify the convergence in unsteady simulations

  • The authors should add the equation of RMS in section 3.2

  • The authors should quantify the uncertainty in the experimental data

  • The authors should improve the figure quality starting from section 3.1. The legend font size in some figures is too small to read.  Additionally, in the point of this reviewer, the authors should use rows instead of columns to re-arrange figures, thus reducing the number of page

  • The authors should correct the reference error displayed in the text

Author Response

Dear anonymous reviewer,

We would like to appreciate your review comments and suggestions. We have carefully revised the manuscript according to your comments. Please find our responses to the comments in the following.

Reviewer 2

This paper presents a numerical investigation of the unsteady flow field within a centrifugal fan, containing a long downstream pipe, at three different inlet flow rates, via detached eddy simulations. Both time-averaged and time-resolved flow characteristics are evaluated, serving as a tool to understand the hydrodynamic unsteady regime.

The authors should implement the following revisions before the paper is accepted to publication:

C1: The authors should move section 2.2 to appendix

A1: Thank you so much for the suggestion. Section 2.2 is about the governing equations of the flow as well as the turbulence models, and it is certainly better to be in the Appendix for interested readers to refer to.

C2: In Figure 2, the authors should include a snapshot of near-wall grid

A2: Thank you for the suggestion. In the revised manuscript, a snapshot of the mesh at the blade trailing edge is presented showing the near-wall grid distribution.

C3: The authors should quantify the convergence in unsteady simulations

A3: Thank you for the comment. We have added the convergence criteria settings of the unsteady simulations and a representative plot of the convergence history in the revised manuscript.

C4: The authors should add the equation of RMS in section 3.2

A4: Thank you for the suggestion. We have added the equation of RMS in section 3.2 so that readers can have a clear understanding of the RMS value indicates only the fluctuations since the mean value is subtracted.

C5: The authors should quantify the uncertainty in the experimental data

A5: Thank you for the comment. In the revised manuscript, we have added the uncertainty of the fan performance test rig within 2%. For the pressure fluctuations, the measurement was repeated 8 times, and the standard deviation of the results was found to be within 5%.

C6: The authors should improve the figure quality starting from section 3.1. The legend font size in some figures is too small to read.  Additionally, in the point of this reviewer, the authors should use rows instead of columns to re-arrange figures, thus reducing the number of page

A6: Thank you for the comment. In the revised manuscript, the resolution of the figures has been improved, and the font size in some plots has been increased. According to the suggestion, we have re-arranged the figures to be in rows, and the manuscript now has be reduced to be 22 pages from the original 27 pages.

C7: The authors should correct the reference error displayed in the text

A7: Thank you for the reminder. In the revised manuscript, we have double checked the reference citations. Throughout the manuscript, we use Endnote to cite the references and Mathtype to cite equations.

Round 2

Reviewer 2 Report

The authors have addressed my comments. Therefore, I recommend this paper to the journal.